# Mapping global distributions, environmental controls, and uncertainties of apparent top- and subsoil organic carbon turnover times

Lei Zhang[1,2], Lin Yang[1], Thomas W. Crowther[3], Constantin M. Zohner[3], Sebastian Doetterl[4], Gerard B.M. Heuvelink[5,6], Alexandre M.J.-C. Wadoux[7], A-Xing Zhu[8], Yue Pu[1], Feixue Shen[1], Haozhi Ma[9], Yibiao Zou[3], Chenghu Zhou[1,10]

[1] School of Geography and Ocean Science, Nanjing University, Nanjing, China
[2] Climate and Ecosystem Science Division, Lawrence Berkeley National Laboratory, Berkeley, CA, USA
[3] Institute of Integrative Biology, ETH Zurich (Swiss Federal Institute of Technology), Zurich, Switzerland
[4] Soil Resources Group, Department of Environmental Systems Science, ETH, Zurich, Switzerland
[5] Soil Geography and Landscape Group, Wageningen University, Wageningen, The Netherlands
[6] ISRIC – World Soil Information, Wageningen, The Netherlands
[7] LISAH, Univ Montpellier, AgroParisTech, INRAE, IRD, L'Institut Agro, Montpellier, France
[8] Department of Geography, University of Wisconsin-Madison, Madison, WI, USA
[9] Swiss Federal Institute for Forest, Snow and Landscape Research WSL, Birmensdorf, Switzerland
[10] State Key Laboratory of Resources and Environmental Information System, Institute of Geographical Sciences and Natural Resources Research, Chinese Academy of Sciences, Beijing, China

*Correspondence to*: Lin Yang (yanglin@nju.edu.cn) and Chenghu Zhou (zhouch@lreis.ac.cn)

**Abstract.** The turnover time (τ) of global soil organic carbon is central to the functioning of terrestrial ecosystems. Yet our spatially-explicit understanding of depth-dependent variations and environmental controls of τ at a global scale remain incomplete. In this study, we combine multiple state-of-the-art observation-based datasets, including over ninety thousand geo-referenced soil profiles, the latest root observations distributed globally, and large amounts of satellite-derived environmental variables, to generate global maps of apparent τ in topsoil (0–0.3 m) and subsoil (0.3–1 m) layers with a spatial resolution of 30 arcsec (~1 km at the Equator). We show that subsoil τ ($385_{20}^{3485}$ years [mean with a variation range from 2.5th to 97.5th percentile]) is over eight times longer than topsoil τ ($15_{11}^{137}$ years). The cross-validation shows that the fitted machine learning models effectively captured the variabilities in τ, with $R^2$ values of 0.87 and 0.70 for topsoil and subsoil τ mapping, respectively. The prediction uncertainties of the τ maps were quantified for better user applications. The environmental controls on top- and subsoil τ were investigated at global, biome, and local scales. Our analyses illustrate that how temperature, water availability, physio-chemical properties and depth exert jointly impacts on τ. The data-driven approaches allow us to identify their interactions, thereby enriching our comprehension of mechanisms driving nonlinear τ–environment relationships from global to local scales. The distributions of dominating factors of τ at local scales were mapped for identifying context-dependent controls on τ across different regions. We further reveal that the current Earth system models may underestimate τ by comparing model-derived maps with our observation-derived τ maps. The resulting maps with new insights demonstrated in this study facilitate the future modelling efforts of carbon cycle–climate feedbacks and supporting effective carbon management. The dataset is archived and freely available at https://doi.org/10.5281/zenodo.14560239 (Zhang, 2025).

# 1 Introduction

As the largest active reservoir of organic carbon in terrestrial ecosystems, soils are integral to the global carbon cycle (Schimel, 1995; Batjes, 1996; Smith et al., 2024). Plants capture $CO_2$ from the atmosphere through photosynthesis and transfer carbon into soils through litter fall and root exudates. This carbon is then cycled back to the atmosphere through heterotrophic respiration by organic-matter decomposers, a process governed by decomposition rates (Balesdent et al., 2018). The turnover time of soil organic carbon (SOC), denoted as $\tau$ (in years), is the average time that organic carbon molecules remain in the soil (Six and Jastrow, 2002; Sierra et al., 2017). This turnover time is a critical factor in determining the size of soil carbon pools (Sierra et al., 2017; Crowther et al., 2019b). Understanding the spatial variation in $\tau$ and the underlying environmental drivers is therefore crucial for comprehending the scale and dynamics of terrestrial carbon storage under current and future climate change scenarios (Torn et al., 1997, 2009; Field et al., 2014).

Several studies have estimated the global apparent carbon turnover times in whole terrestrial ecosystems (Carvalhais et al., 2014; Fan et al., 2020). Nevertheless, the detailed patterns of $\tau$ in soil systems at the global scale remain to be elucidated. Much work has focused on shallow soil horizons as the higher carbon content and the greater availability of data in topsoils (e.g., Crowther et al., 2016; Luo et al., 2017; Viscarra Rossel et al., 2019; Wu et al., 2021). However, capturing the biogeographic variability of carbon dynamics in deeper soil layers is emerging as a critical area of research (Hicks Pries et al., 2023), as the environmental sensitivities of SOC there can substantially differ from surface layers (Rumpel and Kögel-Knabner, 2011; Hicks Pries et al., 2017; Luo et al., 2019; Soong et al., 2021; Zosso et al., 2023). To enhance our understanding of soil carbon turnover and address these issues, there is a clear need for a global analysis of $\tau$ that considers both top- and subsoil layers, and providing spatially-explicit $\tau$ maps. Such an analysis should leverage the latest and most comprehensive global soil profile datasets available, enabling more reliable assessments and insights into terrestrial carbon sink potential.

The ensemble of latest datasets also supports a more nuanced understanding of the environmental controls on $\tau$, particularly for their potentially non-linear or distinct effects across different spatial scales and soil depths. Previous studies showed evidence that the turnover time of SOC is negatively correlated with temperature and precipitation (Davidson and Janssens, 2006; Chen et al., 2013; Wang et al., 2018), and that subsoil carbon may be particularly sensitive to temperature fluctuations (Jia et al., 2019; Soong et al., 2021; Chen et al., 2023). Yet, other studies could not confirm this strong climatic dependency (Giardina and Ryan, 2000; Doetterl et al., 2015), suggesting a predominant influence of soil properties on subsoil carbon turnover times in certain regions or over decadal timescales (Luo et al., 2019). This inconsistency underscores the need for a multifaceted approach to quantify the effects of multiple factors on $\tau$, particularly the interactions between climate and edaphic factors, across different spatial scales and soil depths (Schmidt et al., 2011). Comprehensive assessments at the global, biome, and local levels will be crucial to identify the primary controls of $\tau$ for both topsoil and subsoil layers. Moreover, observation-based global estimates of $\tau$ are essential for simulating the global carbon cycle (Todd-Brown et al., 2013; Friend et al., 2014; Varney et al., 2022). An accurate representation and deeper understanding of the

environmental controls of τ – spanning diverse spatial scales and soil depths – will be integral to benchmarking current Earth system models (ESMs) and reducing bias in future carbon cycle projections.

This study aims to develop a global estimation of τ by integrating the state-of-the-art soil and root profile databases with satellite-derived environmental observations. The collected datasets allow us to firstly estimate τ at over ninety thousand global sampling sites, and then we used machine learning methods to generate a spatially-explicit understanding of global SOC turnover times in the top- (0–0.3 m) and subsoil (0.3–1 m). To comprehend the interactive mechanisms among multiple environmental drivers that have shaped variations in top- and subsoil τ at the global, biome-level and local scales, we used data-driven approaches to characterize the directional contributions of climate, topography, physical and chemical properties of soil to explain τ patterns. We further quantified the uncertainty maps for better user applications, and compared our observation-derived τ with ESM-derived τ in both spatial variabilities and climatic dependencies.

## 2 Materials and methods

### 2.1 Estimation of τ in top- and subsoil layers

The SOC turnover time (τ) we estimated in this study is the mean transit time that the newly entered carbon spends in soils until it leaves (Six and Jastrow, 2002; Sierra et al., 2017). When assuming a steady state and homogeneity of the system in which all particles have the same probability of leaving at any time, τ can be defined as the ratio between SOC stock (SOCS) to input or output flux of carbon (τ = SOCS/flux), which can be called the apparent turnover time (Carvalhais et al., 2014; Fan et al., 2020). We estimated SOC stock (see Section 2.1.1) and vertical allocation of carbon input in two soil layers (see Section 2.1.2), and then combined them to calculate top- and subsoil τ at all soil profile locations. We then adopted the point-level estimates to map τ variation across the globe.

#### 2.1.1 Estimation of SOC stocks based on soil sample databases

Soil sample data were mainly collected from the standardized soil profile database provided by World Soil Information Service (WoSIS) (Batjes et al., 2020). The Northern Circumpolar Soil Carbon Database (NCSCD) (Hugelius et al., 2013) was incorporated to supplement samples across the high latitudes of the Northern Hemisphere. In addition, the soil sample data from the project of National Soil Survey of China was incorporated (Zhang et al., 2013; Liu et al., 2022). To minimize bias due to erroneous and/or uncertain measurements, we removed samples i) with low accuracy of the geographical coordinates (i.e., the information of degree, minute and second is not fully provided in the source), ii) located in areas with exceptionally low NPP (below 10 gC m$^{-2}$ yr$^{-1}$), iii) with low quality of NPP estimates (if the quality control value is larger than 50%), and iv) layer observations flagged as surficial litter. Finally, a total number of 95,200 and 66,807 geo-referenced sampling locations providing topsoil and subsoil information were collected for this study (Fig. 1). The estimation of SOC

stock (SOCS, kgC m$^{-2}$) for each soil sample site at a certain depth interval between the upper depth ($D_u$, m) and lower depth ($D_l$, m) can be computed as follows:

$$SOCS_{D_u-D_l} = SOC_{D_u-D_l} \cdot BD \cdot \left(1 - \frac{CF}{100}\right) \cdot (D_u - D_l) \tag{1}$$

where $SOC_{D_u-D_l}$ is the SOC content (g kg$^{-1}$) in a certain range of soil depths, BD is the soil bulk density (g cm$^{-3}$), and CF is the percentage of coarse fragments in the whole soil (%). We adopted an approach described in Text S1 to fill the missing values by using a specific pedotransfer function depending on the SOC content and the considered layer. When CF was missing, we extracted the corresponding data at their locations from the latest version of SoilGrids maps (Poggio et al., 2021). The equal-area spline algorithm was employed to fit observations along depths (Bishop et al., 1999; Malone et al.,

2009). The average of fitted values was adopted to estimate the SOCS at top- and subsoil layers for each profile (Figs. S1). The probability distributions of SOCS, BD, and CF in different biomes are shown in Figs. S2-S4.

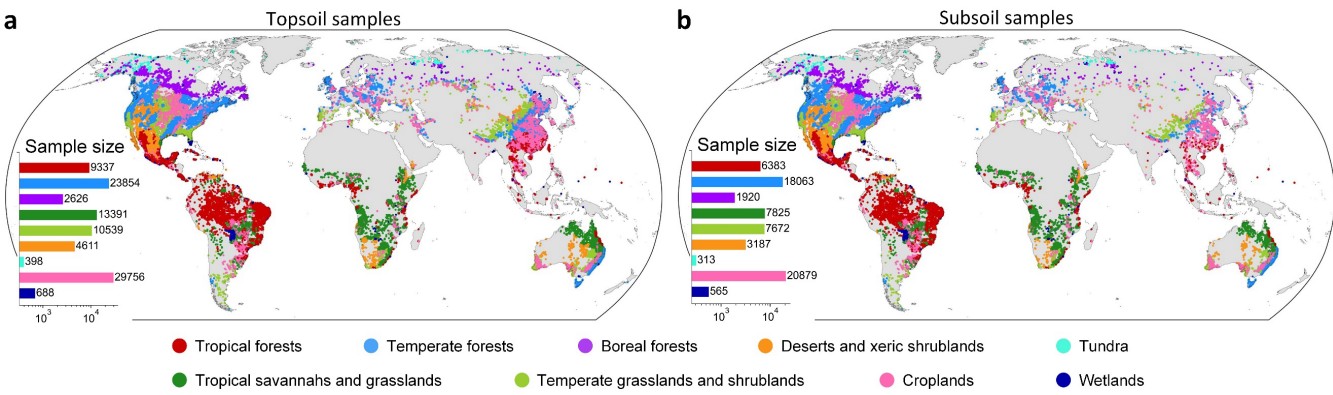

**Figure 1: Spatial distributions of geo-referenced soil samples used in the study across global biomes.** A total of 95,200 (**a**, for topsoil) and 66,807 (**b**, for subsoil) sampling sites were collected from multiple soil profile databases. The bar plots show the sample size within

each biome.

### 2.1.2 Estimation of carbon allocation belowground

Carbon influx at each soil layer comes from vertical allocation of the amount of net primary productivity (NPP). This involves the estimations of above- and belowground NPP and NPP allocation into different soil layers. The annual NPP (kgC m$^{-2}$ yr$^{-1}$) produced by the moderate-resolution imaging spectroradiometer (MODIS) was collected at each sample location

(Running and Zhao, 2019). The MOD17 version 6 product (https://ladsweb.modaps.eosdis.nasa.gov) was used in this study. The mean of annual NPP from 2001 to 2019 was computed for representing the general status of each sample location. Then, we used the biomass ratio between roots and shoots (root-to-shoot ratio, RSR) as a proxy to estimate the amount of NPP allocated to soils, as it is generally a realistic estimation for the mean long-term NPP partitioning (Gower et al., 1999). The fraction of belowground NPP can be estimated as the root-mass fraction (RMF):

$$RMF = \frac{RSR}{RSR + 1} \tag{2}$$

The RSR value at each sample location was collected from the harmonized global maps of above and belowground biomass carbon density (Spawn et al., 2020). The biome-specific variabilities of RMF are shown in Fig. S5.

We used the vertical root biomass distribution to represent the belowground NPP partitioned into different soil depths (Luo et al., 2019; Xiao et al., 2022, 2023). The root distribution information was obtained from the global 564 root profiles complied by Schenk and Jackson (Schenk and Jackson, 2002). The logistic dose–response curve function was used to
estimate the cumulative amount of root mass $r(D)$ above a certain soil depth $D$ (m):

$$r(D) = \frac{R_{max}}{1 + \left(\frac{D}{D_{50}}\right)^c} \tag{3}$$

where $R_{max}$ represents the total root mass, $D_{50}$ is the depth at which $r(D)$ equals to the half of $R_{max}$, and $c$ is a dimensionless shape-parameter which can refer to Schenk and Jackson (2002) for details. The fraction of roots in a certain soil layer between $D_u$ and $D_l$ ($fr_{D_u-D_l}$) can be estimated as follows:

$$fr_{D_u-D_l} = \frac{r_{D_l}}{R_{max}} - \frac{r_{D_u}}{R_{max}} = \frac{1}{1 + \left(\frac{D_l}{D_{50}}\right)^c} - \frac{1}{1 + \left(\frac{D_u}{D_{50}}\right)^c} \tag{4}$$

This root profile dataset has been used to analyze the belowground NPP allocation in several previous studies (Luo et
al., 2019; Shi et al., 2021). In our study, we extended this data by collecting the latest Root Systems of Individual Plants (RSIP) database and the global root traits (GRooT) database that includes the information of the maximum belowground extents of terrestrial plants (Guerrero-Ramírez et al., 2021; Tumber-Dávila et al., 2022). By assuming that the root distribution is proportional to its morphological distribution (Bardgett et al., 2014; Tumber-Dávila et al., 2022), we used the form trait of root, the maximum rooting depth ($D_{max}$), to generate an alternative representation of the root distribution. A
total of 1,732 geo-referenced measurements were collected with description of rooting depth from those two databases. The vertical root distribution can be estimated according to a commonly used asymptotic equation (Gale and Grigal, 1987; Jackson et al., 1996; Zeng, 2001):

$$CRF(D) = 1 - \beta^D \tag{5}$$

where CRF is the cumulative root fraction from the surface to soil depth $D$ in centimeters, $\beta$ is the estimated parameter which controls the decreasing rate of root mass with increasing soil depth (Gale and Grigal, 1987; Zeng, 2001). Finally, the
fraction of roots in a certain soil layer between upper and lower depth ($fr_{D_u-D_l}$) can be quantified as: $CRF(D_l) - CRF(D_u)$. An illustration of the root distribution estimation through the above approach is shown in Fig. S6.

Hence, the global distributions of all root profiles that characterize the fraction of root in two layers were calculated and are shown in Fig. S7. Given that root distribution is generally related with the biome, vegetation type and soil conditions (Jackson et al., 1996; Schenk and Jackson, 2005), we apply the arithmetic mean of $fr$ from root profile observations within the same terrestrial ecoregions (Dinerstein et al., 2017) and soil type (FAO–Unesco, 1990) as soil sample (Figs. S8 and S9).

The vertically physical transportation of organic carbon, such as leaching and/or bioturbation, is also necessary to be considered. We followed the function designed in previous model that includes vertical transport of SOC (Braakhekke et al., 2011; Koven et al., 2013; Sierra et al., 2024):

$$V(D) = Diff \frac{\partial^2 C}{\partial D^2} \tag{6}$$

where $V$ is the transported SOC stock ($V$) in a certain layer; $C$ is the organic carbon content, which is defined volumetrically (kg m$^{-3}$), at the depth $D$; $Diff$ is the diffusivity which is constant and set to be $1\times10^{-4}$ m$^2$ year$^{-1}$ according previous studies (Koven et al., 2013; Sierra et al., 2024). Thus, the belowground NPP (BNPP) in a certain soil layer can be estimated as follows:

$$BNPP_{D_u-D_l} = NPP \cdot RMF \cdot fr_{D_u-D_l} + V_{D_u-D_l} \tag{7}$$

For topsoil, carbon inputs also contain a portion of carbon from surface litterfall, which should be additionally considered. First, the RSR dataset was adopted to support us to obtain the aboveground NPP (ANPP). As a decent fraction of NPP was removed as the harvest products in croplands, we performed a specific calculation procedure to estimate the aboveground carbon input in this region as described in Text S2. Second, we used two databases including measurements of aboveground litterfall across the globe (Holland et al., 2015; Jia et al., 2016), to estimate the fraction of aboveground NPP (ANPP) converted into litterfall. Then, we determined the fraction of litterfall allocated as the carbon input to topsoil according to the decomposition processes described in Community Land Model (Oleson et al., 2013). The details of calculations related with the fraction of ANPP that transferred as carbon input into the topsoil (denoted as $fr_a$) are described in Text S3.

Therefore, the carbon input flux between upper ($D_u$) and lower depth ($D_l$) can be estimated as follows (the biome-specific variabilities of this variable are shown in Fig. S10):

$$flux_{D_u-D_l} = BNPP_{D_u-D_l} + fr_a \cdot ANPP \tag{8}$$

Combining the estimated SOC stock (Eq. 1) and carbon allocation belowground (Eq. 8) with the equation of apparent $\tau$, the values of $\tau$ in top- and subsoil layers were calculated at all sample locations across the globe are shown in Fig. S11. Table S1 shows the details of datasets used to calculate $\tau$ for all samples.

## 2.2 Geospatial mapping of τ

The machine learning-based method was adopted to generate the spatially explicit maps of τ at the two soil layers. As the procedure of geospatial predictive mapping illustrated in Fig. S12, we used the random forest (RF) model to establish the relationship between τ and it potentially related geographical variables (i.e., environmental covariates), for mapping the global distribution of τ with quantification of uncertainty. The advantage of RF is the incorporation of randomized feature selection and training sample selection (Breiman, 2001), so that it can reduce the overfitting risk and lead to a good ability for generalization. This model has been effectively applied on large datasets for geospatial mapping tasks at large scales (van den Hoogen et al., 2019; Ma et al., 2021; Poggio et al., 2021; Guo et al., 2025). There are two important user-defined parameters when using RF model. The first is the number of covariates that randomly selected for each tree building process. We used the rounded down square root of the total number of covariates as this parameter value by default. The second parameter is the number of trees to be learned in RF. We set this value as 200, considering the previous soil mapping studies showed that it is sufficient to obtain stable results when the number of trees is larger than 150 (Wadoux, 2019; Zhang et al., 2021). The covariates including climate, soil physical, soil chemical properties and topography were collected (Table S2 and Figs. S13-S18). These covariate maps were overlaid and thus allowed us to predict τ at the global scale with a spatial resolution of 30 arc-seconds (~1 km$^2$ at the Equator). Considering some regions covered by organic soils cannot be well represented when using our method to generate estimations, therefore, we masked these regions according to the definition of organic soil in Brady & Weil (1999).

## 2.3. Analysis of environmental controls

To investigate the environmental controls of the spatial distribution of τ at two layers, the RF model was adopted based on all soil sample data to quantify the relative importance (*RI*) of each covariate at the global scale. We used permutation-based feature importance as the metric to assess *RI* (Altmann et al., 2010) (see Text S4).

The directional effects of important environmental factors on τ was investigated by using the partial regression model to calculate the partial correlations of top- and subsoil τ with the six top ranked environmental variables from RF modelling mentioned above. The analyzed variables are mean annual temperature (MAT) and precipitation (MAP), the fine particle-size fraction (CLAY+SILT), organic carbon to total nitrogen ratio (C:N), cation exchange capacity of the soil (CEC), and soil pH. The partial correlation of each influencing factor was calculated at the mean level while controlling other factors.

Given the potentially non-significant effects of a variable (e.g. the global impact of precipitation) on τ in a global-level linear model, it does not necessarily imply that the respective variable has no influence on τ. Instead, this lack of significance may arise from its divergent effects when interacting with other factors. This led to an interest in exploring the reasons behind such nonlinear and uncoherent driving mechanisms. Considering the extrinsic climate effects on soil carbon turnover can be interacted by regionally intrinsic soil characteristics (Doetterl et al., 2015), we used an approach (see Text S5) to exemplify the interactive effects of MAT and MAP on top- and subsoil τ in response to the changes of other factors (namely

interactive factors). In addition to the overall importance of covariates analyzed at the large scale, the local-level importance for four categories of variables was also assessed (see Text S6).

## 2.4 Evaluation of mapping results

### 2.4.1 Assessment of accuracy by cross-validation

The mapping accuracy was assessed by using a 10-fold cross validation. The total sample data were divided into ten equally sized subsets, and samples of each biome in each subset has the same proportion as the whole dataset. Nine folds were used as the training data to fit the model, and the prediction was validated on the one remaining fold. This procedure was carried out ten times, and each time using a different fold for validation. The mean value of coefficient-of-determination ($R^2$) and root mean squared error (RMSE) for all folds were computed as the final accuracy metrics for the assessment of mapping results.

### 2.4.2 Quantification of uncertainty

To evaluate the uncertainty of predictive maps of $\tau$, we considered two sources of error. One is the model error, which refers to the covariates that do not fully explain the variations of $\tau$ and the error in the estimation of the model parameters. Another is the measurement error, which represents difference between the actual and recorded value of input variables related to $\tau$ calculation. To account the modelling uncertainty, we adopted quantile regression forests (QRF) (Meinshausen, 2006; Hengl et al., 2018) to derive prediction intervals. QRF first constructs a RF model in the usual way, by developing multiple decision trees that use subsets of the training data, whereby the prediction of each tree equals the average of the observations in the end node of the tree in which the prediction point sits. The RF prediction is the average of all tree predictions. Since averaging is a linear process, the RF prediction boils down to a weighted sum over the $n$ observations of the response variable:

$$\hat{y}(x) = \sum_{i=1}^{n} w_i(x) \cdot y_i \tag{9}$$

In QRF, the weights $w_i(x)$ are used to estimate the cumulative distribution $F(y|x) = P(Y \leq y|x)$ of the response variable $Y$, given the covariate data $x$, as follows:

$$\hat{F}(y|x) = \sum_{i=1}^{n} w_i(x) \cdot 1_{\{y_i \leq y\}} \tag{10}$$

where $1_{\{y_i \leq y\}}$ is the indicator function (i.e., it is 1 if the condition is true and 0 otherwise). Any quantile $q$ of the distribution can then be derived by iterating towards the value of $y$ for which $\hat{F}(y|x) = q$.

We used the R package ranger (Wright and Ziegler, 2017) with the function *quantreg* to build QRF models. Using this function, not only the prediction values at each location can be obtained, but also the 0.05 quantile ($q_{0.05}$), 0.50 quantile

($q_{0.50}$) and 0.95 quantile ($q_{0.95}$) can be computed to derive the lower limit, median and upper limit of a symmetric 90% prediction interval. This interval has also been adopted for uncertainty assessment in GlobalSoilMap specifications (Arrouays et al., 2014) and SoilGrids product (Poggio et al., 2021).

The uncertainty from the input data in calculation of $\tau$ was also considered. When calculating $\tau$ for each sample, four input variables (SOCS, NPP, RMF and $fr_{D_u-D_l}$) introduced uncertainty due to errors in soil measurements and estimation of 230 carbon allocation belowground. Errors in these input variables will propagate to the output of $\tau$ estimation. To incorporate the error propagation from these inputs to the uncertainty evaluation, the standard deviation (SD) of each input at each sample location needs to be quantified. Here, as the calculation of $\tau$ is a simple arithmetic function, the SD of the estimated $\tau$ for each sample can be calculated as follows, when ignoring the cross-correlation among those inputs (Heuvelink, 1998):

$$SD_f = \sqrt{\left(\frac{\partial f}{\partial I_1}\right)^2 SD_{I_1}{}^2 + \left(\frac{\partial f}{\partial I_2}\right)^2 SD_{I_2}{}^2 + \cdots + \left(\frac{\partial f}{\partial I_m}\right)^2 SD_{I_m}{}^2} \tag{11}$$

where $SD_f$ represents the SD of the output value calculated by the function $f$; $I_1$, $I_2$ and $I_m$ represent the first, second, and 235 $m$-th input variables, respectively. For RMF, we directly used the uncertainty maps provided by ref. (Spawn et al., 2020). The SD of SOCS was obtained from Poggio et al. when SOCS was supplemented by extracting values from SoilGrids maps. For quantifying the SD of $fr_{D_u-D_l}$, the SD of all selected root profile observation values for each soil sample site was adopted to represent its uncertainty. For NPP, considering that most MODIS NPP values are within mean ±1SD of the respective values observed from the flux-tower (Reichstein et al., 2002), from which we define the uncertainty of NPP 240 values. The relationships between the estimated values of $\tau$ and their corresponding SD for all samples are shown in Fig. S19. After quantifying the SD of the estimated $\tau$ at each sample location, a Monte Carlo approach was adopted to incorporate the SD in the estimated $\tau$ at each sample. That is, the value of $\tau$ at each sample location was randomly drawn 100 times from a normal distribution given the known mean and SD. Then, all generated samples were used to fit a QRF model and produce the uncertainty maps.

To visualize the spatial distribution of the prediction uncertainty, we calculated the prediction interval ratio (PIR) defined as the ratio of the range between lower and upper limits over the median:

$$PIR = \frac{q_{0.95} - q_{0.05}}{q_{0.50}} \tag{12}$$

A lower PIR indicates higher confidence in the predictions, while a higher PIR suggests greater uncertainty.

We further assessed the quality of the estimated prediction uncertainty through an accuracy plot approach by calculating the prediction interval coverage probability (PICP) (Goovaerts, 2001). PICP is a metric used for evaluating the reliability of 250 uncertainty quantifications by measuring the proportion of observed values that fall within a given prediction interval. It can

effectively evaluate if the probability assigned to the prediction intervals is equal to the frequency of empirical test data within the prediction intervals (Malone et al., 2011; Schmidinger and Heuvelink, 2023).

## 3 Results and discussion

### 3.1 Accuracy assessment

The cross-validation demonstrated that the machine learning models can effectively capture a substantial proportion of $\tau$ variations, achieving $R^2$ values of 0.87 and 0.70, and RMSE values of 14.60 yr and 175.05 yr for the top- and subsoil $\tau$ predictions, respectively (Fig. 2). More accuracy metrics are shown in Table S3. The relatively higher model performance was found in boreal and tundra areas for topsoil and in grasslands and shrublands in warm regions for subsoil, while low model performance was found in temperate forests and subsoil layer in cropland and wetland areas (Figs. S20 and S21). This

is probably because soils in those regions are among the most diverse soil landscapes, compared to pedogenetically similar and more climate driven soils in high latitude regions, and more weathered homogenous soils dominating many tropical lowland areas.

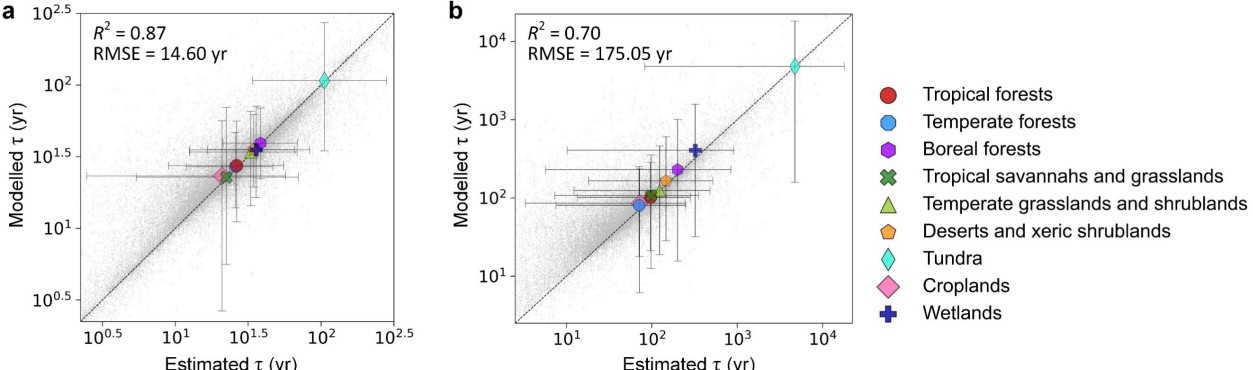

**Figure 2: Validation plots for predictions of soil organic carbon turnover time ($\tau$, yr) in top- (a) and subsoil (b) layer.** Predictions
were generated using random forest regression models. Grey dots are the estimated $\tau$ values based on observations at soil profiles and the predicted values for all validation samples using 10-fold cross-validation. Colored points represent biome-level validation plot. Error bars show 95% percentile intervals of $\tau$ in each biome. Black lines indicate the regression lines between predicted and measured values. Axes are $\log_{10}$-transformed to account for high skewness of $\tau$.

### 3.2 Global distributions of $\tau$ in top- and subsoil

The spatial-continuous maps of global top- and subsoil $\tau$ are shown in Fig. 3. On average, the global $\tau$ in the topsoil was 45 yr, ranging from 11 yr (2.5th percentiles) to 137 yr (97.5th percentile). The average subsoil $\tau$ was 385 yr, ranging from 20 yr to 3485 yr (Table 1). On a global scale, subsoil SOC turnover time was over eight times longer than that in topsoil. The values of $\tau$ generally increased from low to high latitudes, and this latitudinal pattern was more pronounced in the Northern than in the Southern Hemisphere, and it is more evident in the subsoil than in the topsoil layer (Fig. 3b,d). In tropical forests,

the average turnover times were shortest, with approximately 25 yr and 74 yr in the top- and subsoil, respectively. The

turnover times in temperate, desert and cropland areas for top- and subsoil layers were around 29–43 yr and 78–170 yr, respectively. The longest turnover times were found for tundra regions, with an average longer than 90 yr and 1900 yr for the top- and subsoil layers, respectively. Boreal forests and wetlands also show long carbon turnover times, with an average value of over 50 yr and 500 yr, respectively. As such, the differences in average turnover times between the warmest and coldest biomes were more than 60 years and 1800 years for top- and subsoil layers, respectively.

**Table 1. Global- and biome-level statistics of the soil organic carbon turnover time ($\tau$, yr) in top- (0–0.3 m) and subsoil (0.3–1 m) layers.**

| Global/Biome type | Topsoil $\tau$ | | | Subsoil $\tau$ | | |
|---|---|---|---|---|---|---|
| | P 2.5 | Mean | P 97.5 | P 2.5 | Mean | P 97.5 |
| Global | 11 | 45 | 137 | 20 | 385 | 3485 |
| Tropical forests | 12 | 25 | 46 | 27 | 74 | 155 |
| Temperate forests | 15 | 36 | 69 | 17 | 95 | 322 |
| Boreal forests | 27 | 58 | 123 | 47 | 530 | 2569 |
| Tropical savannahs and grasslands | 8 | 31 | 116 | 14 | 101 | 628 |
| Temperate grasslands and shrublands | 10 | 43 | 130 | 18 | 120 | 403 |
| Deserts and xeric shrublands | 16 | 43 | 100 | 30 | 170 | 1250 |
| Tundra | 44 | 91 | 220 | 117 | 1920 | 7505 |
| Croplands | 4 | 29 | 64 | 8 | 78 | 204 |
| Wetlands | 12 | 50 | 187 | 22 | 510 | 5304 |
| Non-permafrost | 9 | 33 | 82 | 17 | 98 | 377 |
| Permafrost | 33 | 75 | 188 | 63 | 1111 | 5836 |

Note: P 2.5 and P 97.5 represent the range of $\tau$ in each biome between the 2.5[th] and 97.5[th] percentiles from the aggregation of all estimated $\tau$ at pixel-level.

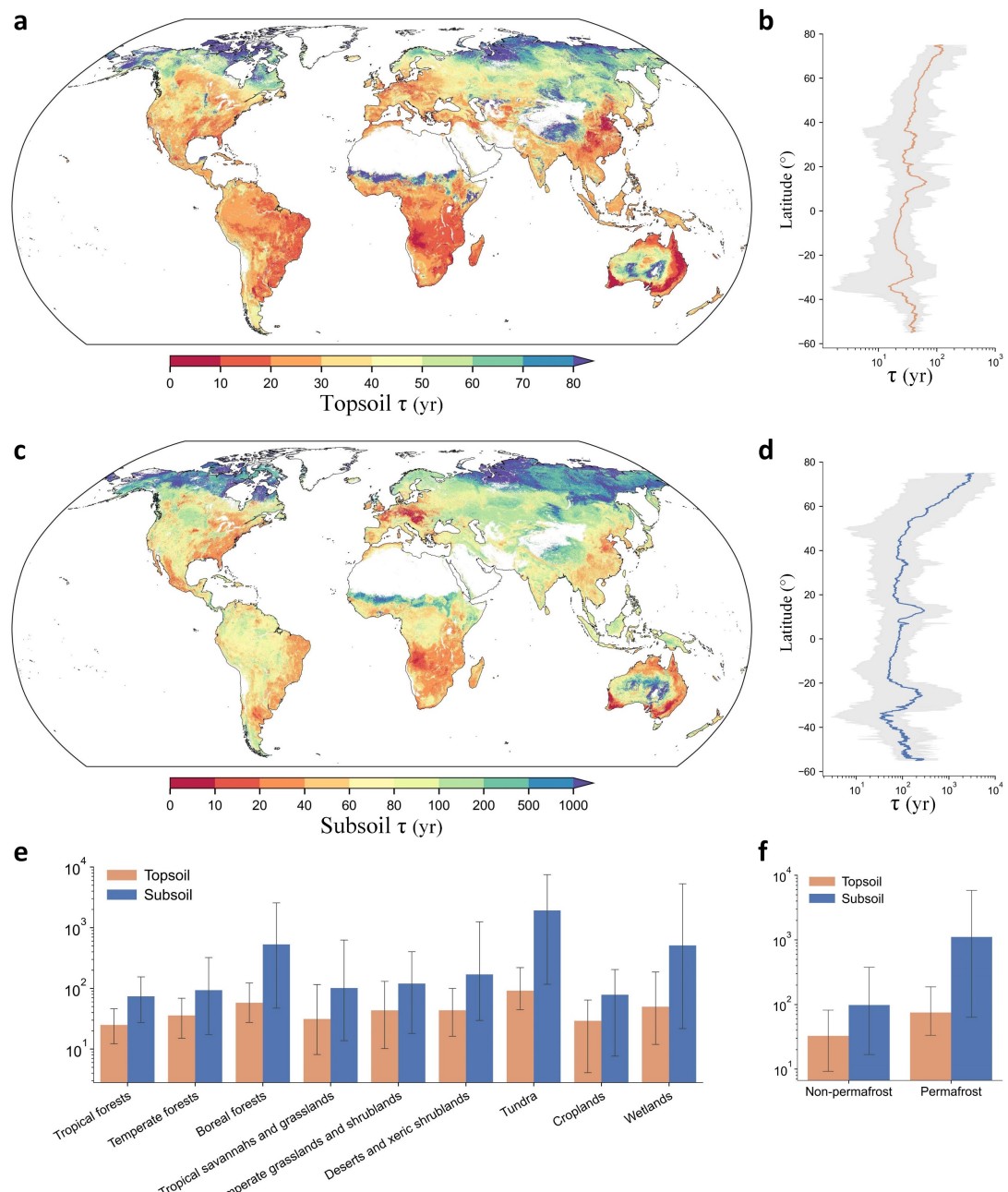

**Figure 3: Global patterns of top- and subsoil organic carbon turnover times (τ, yr).** Global distributions of τ at top- (0–0.3 m) (**a**) and subsoil (0.3–1 m) (**c**) layers. The predicted maps with a spatial resolution of 30 arcsec (~1 km at the Equator) were generated τ-environment relationships using a machine learning model trained from global soil profile observations and their environmental covariates. **b**, **d**, Latitudinal patterns of top- and subsoil τ. Orange and blue lines represent the average τ at top- and subsoil over latitudes, respectively. The shaded grey areas represent the variations of τ between 2.5th and 97.5th percentiles along latitudes. **e**, **f**, Average τ at two layers in different main biomes. Error bars show the 95% percentile intervals of the spatial predictions within each biome.

The uncertainty maps quantified for the top- and subsoil are shown in Fig. 4. Uncertainties of subsoil τ predictions were generally higher than that of topsoil τ, likely due to the fewer horizontal observations in the deep layers. Wide prediction intervals were mostly found in areas with low sampling density, such as deserts and permafrost regions. According to the PICP calculation results, the accuracy plots (Fig. S22) show that the proportion of observed τ values in validation set covered in a certain prediction interval approximately equals the size of that probability interval, which validates the unbiased quantification of uncertainty.

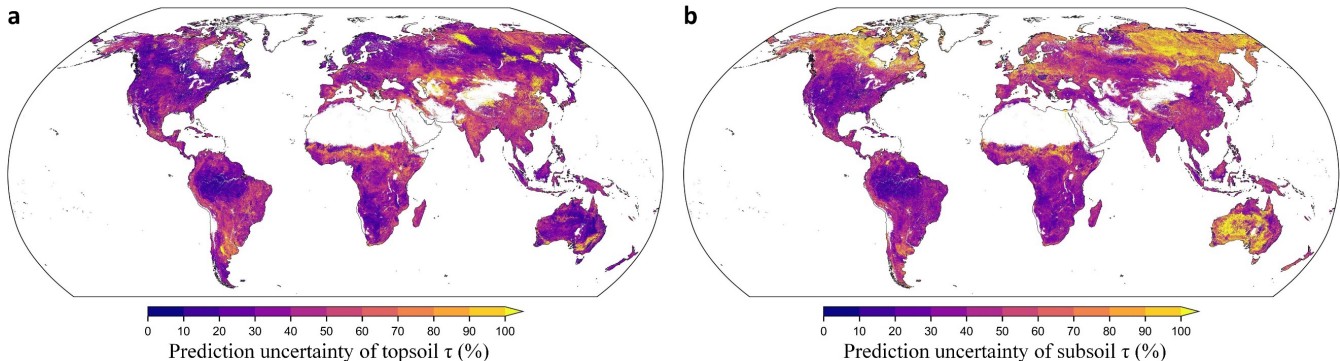

**Figure 4: Uncertainty maps of predicted soil organic carbon turnover time (τ) in top- (a) and subsoil (b) layer.** The uncertainty was quantified by using quantile regression forests. The values shown in the maps represent the prediction interval ratio (PIR), which is the ratio of the range between lower and upper limit (90% prediction interval bounded by the lower [0.05 quantile] and upper [0.95 quantile] limits) over the median (0.50 quantile) of the predictions.

### 3.3 Environmental controls of top- and subsoil τ

### 3.3.1 Global- and biome-level analyses

The relative importance and directional effects of climate, soil (divided into physical and chemical properties) and topographic factors on τ variation at global and biome scales were analyzed (Fig. 5 and Figs. S23, S24). The results suggest scale- and depth-dependency in the drivers of soil turnover times. While climate was found to be the primary driver (explaining nearly half of the total explained variation) of topsoil τ at the global scale, the intrinsic soil characteristics (61%) were more important than climate (32%) for the subsoil τ (Figs. S23a and S24a). This supports recent studies suggesting that soil turnover in deeper layers is less controlled by climatic factors (Luo et al., 2019; Chen et al., 2021; Han et al., 2022) and more by soil properties (Mathieu et al., 2015; Luo et al., 2019). Importantly, by separating soil factors into soil physical (represented by the fine particle-size fraction [CLAY+SILT]) and chemical categories (including C:N, CEC, and soil pH), our study revealed that the topsoil τ is more influenced by soil physical properties than chemical properties (33% *versus* 15%). However, for subsoil τ, soil physical properties are less important than soil chemical properties (27% *versus* 39%). These results support that, in addition to the direct control of climate, pedologic traits and geochemistry of soils exert equivalent or even stronger control on τ at the global scale (Davidson and Janssens, 2006; Doetterl et al., 2015, 2016).

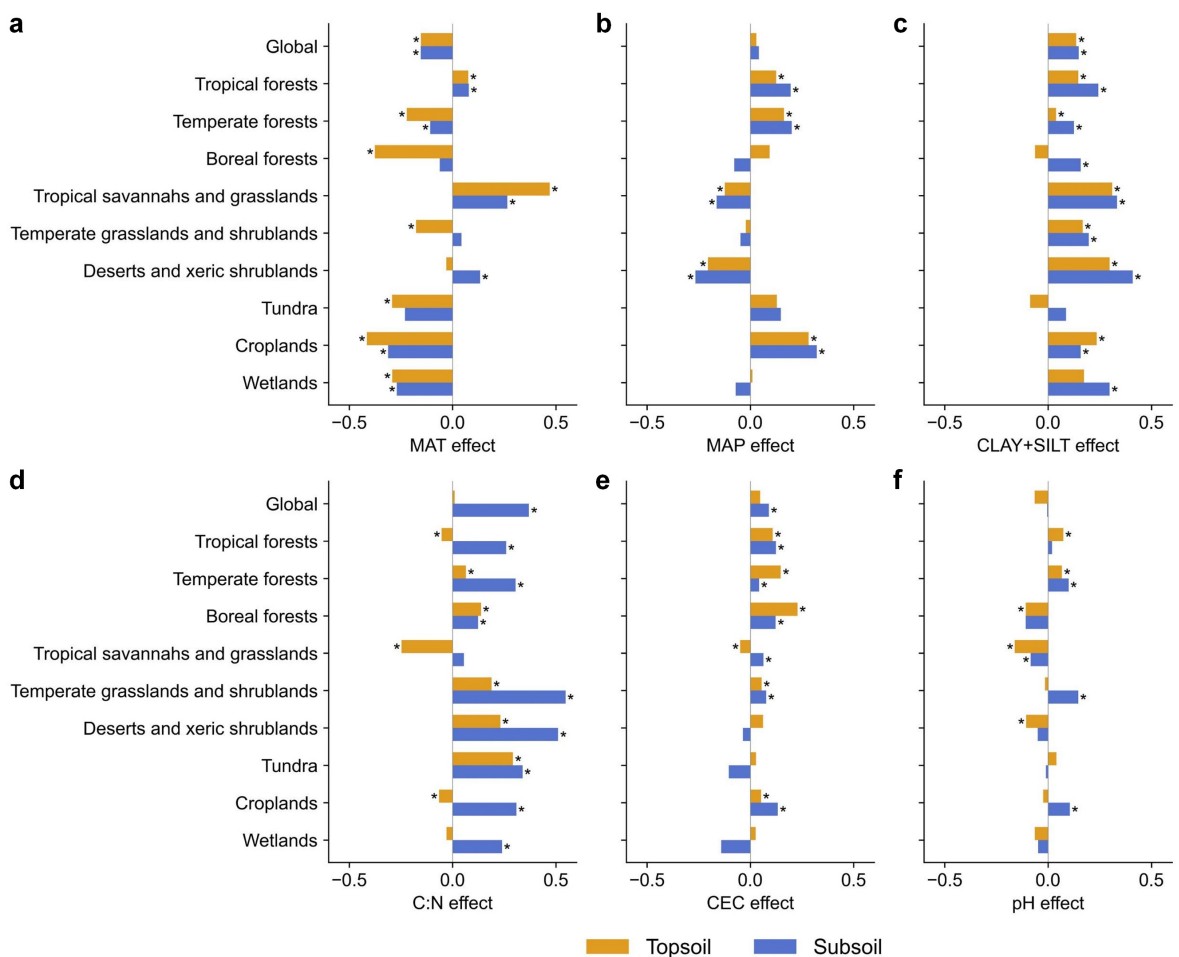

**Figure 5: Relationships between environmental variables and soil organic carbon turnover time (τ) in top- and subsoil layers.** Effects of six important variables on τ are shown at global and biome levels. Values on bar plots represent coefficients between τ and each variable from partial-regression model, where positive or negative values indicate a positive or negative effect on τ, respectively. MAT: mean annual temperature; MAP: mean annual precipitation; CLAY+SILT: fine particle-size fraction (the sum of clay and silt content); C:N: organic carbon to total nitrogen ratio; CEC: cation exchange capacity of the soil; pH: soil pH. *$P < 0.001$.

In different biomes and soil depths, the effects of MAT and MAP show different magnitudes or even opposite directions (Fig. 5a,b). Although a general negative effect of MAT on τ has been found in a previous meta-analysis (Chen et al., 2013), our results show that such effect was reduced and even changed to be positive in tropical regions (Fig. 5a). This can be attributed to the contrasting in relationship of NPP with temperature between tropical and extratropical regions, which stem from their distinct limiting factors affecting plant growth (Fig. S25) (Chapin et al., 2011; Slot and Winter, 2016). In addition, the effect of MAP on τ was not significantly linear at the global scale. This is inconsistent with a previously detected negative relationship (Schimel et al., 1994; Chen et al., 2013), but supports a recent global study that detected a nonlinear relationship between τ and hydrometeorological conditions (Fan et al., 2022). Such previous discrepancies can be explained by contrasting the effects of MAP across different biomes (Fig. 5b). While warm forests and croplands exhibit significant

positive effects, the notable negative effects are shown in tropical savannahs and grasslands, and arid regions. Additionally, it shows MAP negatively impacted subsoil $\tau$ in both boreal forests and temperate grassland and shrublands, but this effect was not significant for topsoil. These $\tau$–climate patterns highlight the nonlinear effects of temperature and water availability on $\tau$ globally and the different driving mechanisms across biomes.

Our analyses further revealed that soil physical and chemical properties have comparable or greater effects on $\tau$ than climatic effects at global and biome scales (Fig. 5c-f). For soil physical properties as represented by the CLAY+SILT, the fraction of fine particle-size in soils was positively related with $\tau$ in general (Fig. 5c). This is mainly because that soils dominated by finer particles tend to stabilize SOC by physical and organo-mineral stabilization mechanisms, while the well-drained sandy soils generally providing limited protection of SOC against microbial decomposition (Krull et al., 2003;

Lützow et al., 2006; Cotrufo et al., 2019). The relatively higher effects of CLAY+SILT on subsoil $\tau$ than that on topsoil $\tau$ (except for croplands) further indicate that mineral-associated and physically protected SOC in deep layer may plays a crucial role in stabilizing SOC stocks, and consequently decrease its climate sensitivity (Gillabel et al., 2010; Qin et al., 2019).

        Among the effects of soil chemical properties, the C:N ratio plays a key role in regulating subsoil $\tau$, ranking as the most

influential soil factor (Fig. S24a). The positive impact of C:N was observed in all biomes for subsoil and specifically in boreal forests, temperate grasslands, deserts, and tundra for topsoil (Fig. 5d). However, the effects of C:N were not found significant for topsoil $\tau$, and it even shows a negative impact in tropical and cropland regions. This finding implies that, in general, nutrient limitations decrease soil respiration and extend carbon residence time by favouring the establishment of slow-decomposing organisms or reducing organic matter quality (Crowther et al., 2019b; Li et al., 2012; Zhou et al., 2007).

Whilst this effect is pronounced in the subsoil layer, it cannot be straightforwardly generalized to the topsoil in tropics and human-impacted regions. The factor of CEC showed a generally positive impact on $\tau$ in forests and croplands, but its effect was weaker in other biomes (Fig. 5e). Soil pH shows a non-significant linear correlation with $\tau$ globally, yet it exhibits divergent influences across different biomes and soil depths (Fig. 5f). It has a positive impact on $\tau$ in warm forests and on subsoil $\tau$ in temperate grasslands and croplands. Conversely, in boreal forests, tropical grasslands, and arid regions, it more

negatively impacts $\tau$ in topsoil.

        In addition to the climate and soil factors, topography can influence $\tau$ by affecting soil moisture distribution, drainage conditions, and permafrost dynamics. Our analysis (Figs. S23h, S24h) indicates that topographic variables contribute significantly to $\tau$ variability in tundra regions. This might be related to the previous studies found that terrain-driven hydrological differences strongly influence permafrost carbon stability (Schuur et al., 2008; Mishra et al., 2021). This also

aligns with research showing that microtopographic variations, such as polygonal tundra structures, create distinct carbon accumulation and decomposition regimes, further influencing SOC turnover times (Jorgenson et al., 2010; Liljedahl et al., 2016).

### 3.3.2 Interactive effects of climatic and edaphic factors on τ

The detected relationships between environmental factors and τ prompted us to delve deeper into the nonlinear driving
mechanisms. We aimed to uncover potential interactions among the primary climatic and edaphic drivers that might be
masked by analyzing the directional effects of only a single variable. We illustrate how fluctuations in one climatic variable
(MAT or MAP) related with changes in another climatic variable (Fig. 6a,f) and four key soil properties (Fig. 6b-e, g-j). We
found that the negative effect of MAT on τ was magnified with decreasing MAP (when MAP < 1500 mm). Meanwhile, it
also shows that MAT effect on τ changed to be smaller, even can be positive for subsoil, with increasing MAP (when MAP
> 2000 mm) (Fig. 6a). Notably, the MAP effect transitioned from negative to positive with increasing MAT from colder
environments. Its effect then diminished after a critical point around 15°C (Fig. 6f).

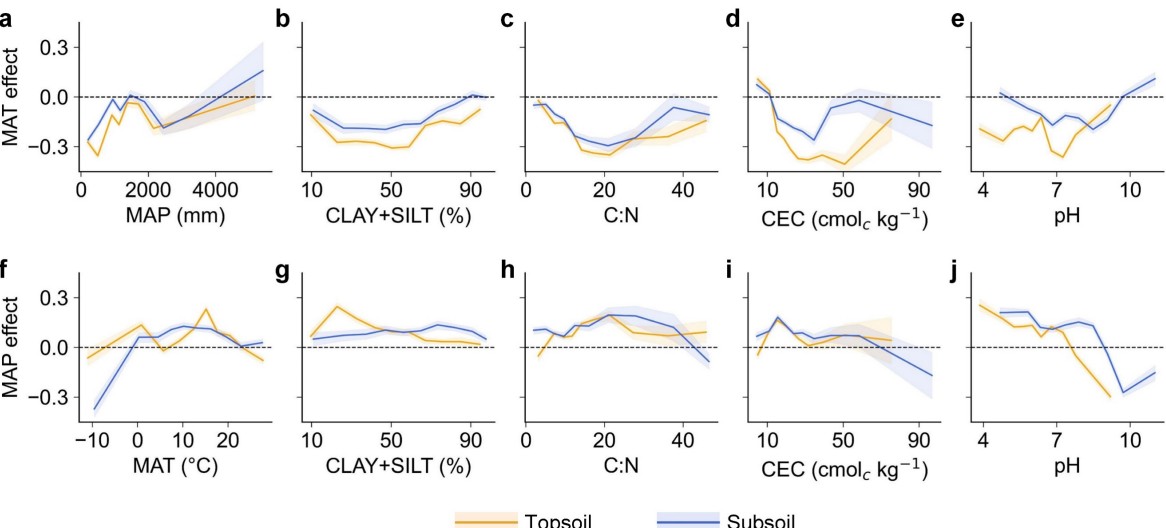

**Figure 6: The climatic effects on top- and subsoil organic carbon turnover time (τ) interacted by other factors.** The lines show the
mean annual temperature (MAT) effect and mean annual precipitation (MAP) effect on τ in response to another climatic variable and four
important soil physio-chemical (including the fine particle-size fraction [CLAY+SILT], carbon-to-nitrogen ratio [C:N], cation exchange
capacity [CEC] and soil pH) variables. The variables on the x-axis can be considered as "interactive factors", which influence the MAT
and MAP effect on τ. The values on the lines above or below dashed horizontal lines indicate the positive or negative climatic effect on
top- or subsoil τ (the MAT/MAP effect was calculated by the partial regression coefficient between MAT/MAP and τ) in response to the
corresponding values of interactive factors. The shaded areas on the lines represent confidence intervals for the results of partial
correlation analysis.

Beyond the interplay of climatic factors, soil property values significantly modulated the effects of MAT and MAP on
τ. For instance, as CLAY+SILT content decreased, the negative impact of MAT was pronounced (Fig. 6b). The MAP effect
was also accentuated in sandy soils but it is restricted to the topsoil layer (Fig. 6g). This result suggests that the finer soil
texture may reduce carbon turnover sensitivity to temperature or water fluctuations (Krull et al., 2003). Among the
interactions of soil chemical properties with climatic impacts on τ, it is evident that extremely low or high C:N ratios in soils
lead to smaller climatic effects on τ (Fig. 6c,h), suggesting that the sensitivity of τ to climate is strongly influenced by
whether the C:N ratio falls within an optimal range. The negative impacts of temperature was intensified with rising CEC

(when CEC < 40 cmol$_c$ kg$^{-1}$) (Fig. 6d), mainly because higher CEC typically enhances the nutrient-supplying capacity of soils, thus quickening carbon turnover in warm conditions (Crowther et al., 2019a). The different responses of topsoil and subsoil temperature sensitivity to CEC (Fig. 6d) may result from interactive and combined influencing factors. Higher CEC enhances nutrient availability for microbes and promotes mineral-organic compound formation, potentially increasing SOC temperature sensitivity. Concurrently, elevated CEC can promote plant productivity, leading to greater carbon inputs and influencing NPP sensitivity to temperature. The observed patterns likely reflect the combined effects of these processes. The influence of soil pH on MAT effects on τ demonstrated a strong nonlinear trend. A more pronounced negative MAT effect was found when soil pH is neutral for topsoil and mildly alkaline for subsoil (Fig. 6e). This is related to that microorganism has a physiologic optimum pH (Rousk et al., 2010; Don et al., 2017), and results in the warmer temperature significantly accelerate SOC turnover rate when pH is within an optimum range (Frostegård et al., 2022; Xiang et al., 2023). Nevertheless, there is currently a need to further explore reasons explaining these pH ranges are different with soil depth, as well as its species-specificity (Bahram et al., 2018). The impact of MAP appears to be more positive in acidic soils, but becomes negative when soil pH exceeds 8 and 9 for topsoil and subsoil, respectively (Fig. 6j). The positive MAP effect on turnover time under acidic conditions could be partly explained by the enhanced effect of increased moisture on soil weathering processes (Porras et al., 2017). Under alkaline conditions, which are mostly in dry climates (Slessarev et al., 2016), the negative MAP effect may arise from the intensified microbial transformation of plant-derived organic matter as these soils provide the favorable pH and moisture to microorganisms (Yang et al., 2022).

### 3.3.3 Distributions of dominating factors of τ at local scales over the world

We further analyzed the effects of these covariates by generating τ–environment models from data at sampling locations within different local areas to assess the deviation of regional patterns from that in the large scale. Overall, we find that the dominant factors controlling τ vary across different local areas (Fig. 7a,b). Across approximately 41% of the land surface area, climatic factors are the predominant controller of topsoil τ (Table 2). However, for subsoil τ, climatic factors predominantly mediate the variation of τ in only 29% of global areas. It is noteworthy that, in 57% of the global area, soil characteristics were the dominant driver of subsoil τ. Overall, soil chemical properties had a larger effect than soil physical properties. It is particularly notable in parts of tropical forests of the Amazon, the broadleaf forests spanning from eastern Indian to southeast Asia, the most parts of North America, central European, and the grasslands, deserts and xeric shrublands scattered across Africa and Australia (Fig. 7b). This phenomenon is related with previous studies suggesting that nutrient availability for soils and vegetation greatly affects τ (Cleveland and Townsend, 2006; Carvalhais et al., 2014). The contributions of four categories of environmental factors from the local-scale analysis differed less in subsoil compared to that in topsoil (Fig. 7c,d and Table 2).

**Table 2. Statistical summary of the local-scale analytical results of variable (classified into four categories) importance on influencing top- and subsoil organic carbon turnover time (τ) at the local scales.**

| Category of variables influencing $\tau$ | Mean importance of a certain category of variables influencing $\tau$ at the local scale | | Percentage of global areas that $\tau$ dominated by a certain category of variables at the local scale | |
|---|---|---|---|---|
| | Topsoil | Subsoil | Topsoil | Subsoil |
| Climate | 0.37 | 0.20 | 41% | 29% |
| Soil physical properties | 0.21 | 0.29 | 18% | 16% |
| Soil chemical properties | 0.28 | 0.17 | 29% | 41% |
| Topography | 0.17 | 0.18 | 12% | 14% |

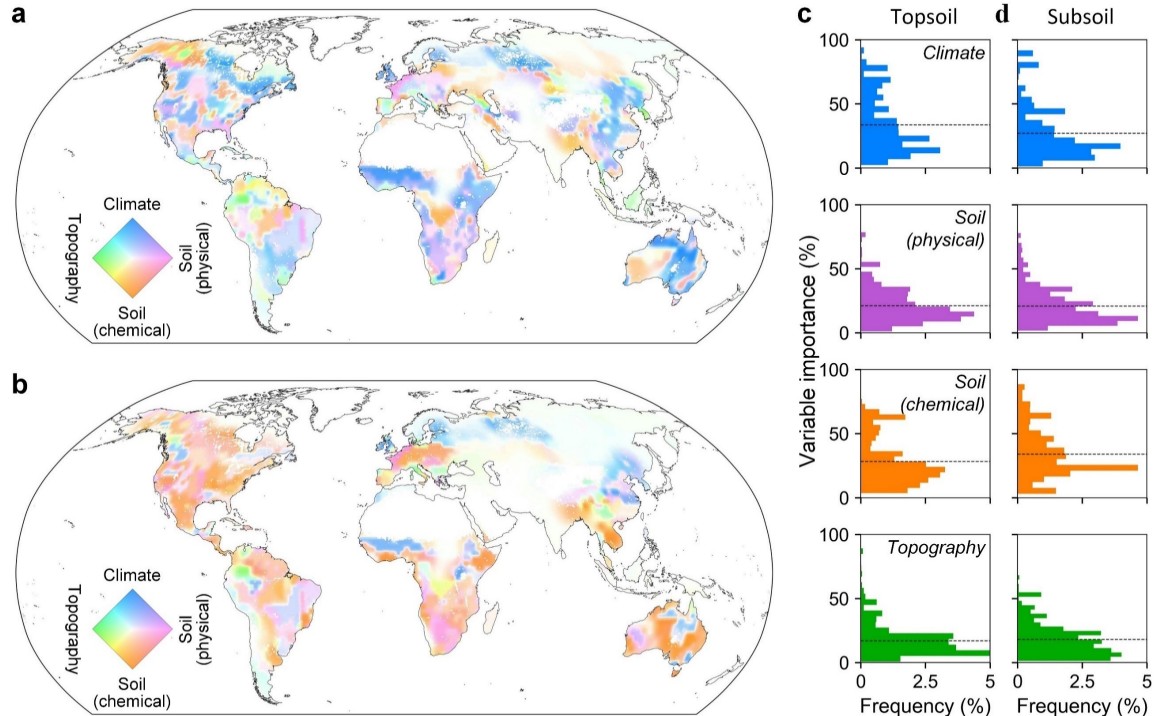

**Figure 7: Local-level analysis of dominating factors of top- and subsoil organic carbon turnover time ($\tau$).** Global maps of dominant factors controlling $\tau$ in top- (**a**) and subsoil (**b**) at the local scales. Maps were created by interpolating all results of factor importance from modelling using the sample data within each local area stratified by the ecoregion map. The lighter color would be displayed if the reliability of the local result is low with a smaller number of available samples for analysis. Histograms of the local-level percentage contribution of each variable category in controlling top- (**c**) and subsoil (**d**) $\tau$ across all local areas, respectively. Dashed lines represent the mean values.

## 3.4 Comparison with $\tau$ from Earth system models

Our observation-based $\tau$ estimates can help constrain ESM simulations and improve predictions of current and future carbon cycle dynamics. We analyzed historical simulation outputs of selected ESMs from CMIP6 (Table S4). While the comparison of our empirical estimates with $\tau$ estimates from ESM outputs based on CMIP6 showed broad agreement in the spatial

patterns of τ (Pearson'*r* = 0.53, *P* < 0.001), it highlighted that ESMs likely underestimate τ across the majority of the globe (Fig. 8a). The soil carbon turnover times estimated from ESMs were on average more than two times shorter than our data-derived (using ground-sourced samples integrated with remote sensing observations) τ estimates (Table S4). This discrepancy was particularly pronounced in tropical forests, grasslands and tundra areas (Table S5). In ~30% of the global land grids, the τ estimates from ESMs did not fall into the 90% estimation intervals of our data-driven estimates, and ~92% of global land area with a mean underestimation bias (ESM-derived τ < data-derived τ) (Fig. 8a, Table S4).

The discrepancies between τ from observations and from ESMs are also reflected in their associations with climate variables. The ESMs predicted stronger correlations with temperature and precipitation compared to our τ estimates (Fig. 8b,c). Additionally, there were notable differences in the climate correlations between topsoil and subsoil, which ESMs do not yet capture. These findings emphasize that other factors, such as soil physico-chemical properties and soil depth, need to be accounted for in models to accurately project global soil carbon dynamics.

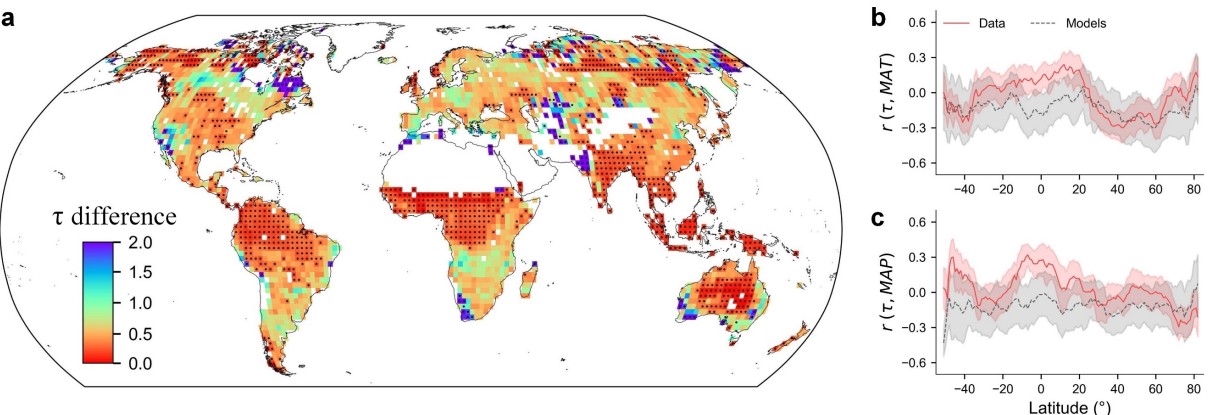

**Figure 8: Comparisons of soil organic carbon turnover times (τ) from our data-derived estimates and from Earth system models (ESMs). a**, Biases in τ represented by ratios of between data-derived to model-derived values across the global grid cells. Black points indicate locations where τ from ESMs are outside the 0.05 and 0.95 quantiles from the prediction uncertainty in our data-derived estimations. **b**, **c**, Comparisons of associations of data- and models (ESMs)-derived τ to mean annual temperature (MAT) (**b**) and mean annual precipitation (MAP) (**c**) along latitude gradients. The *y*-axis shows partial correlations (*r*) of τ with two climate factors, controlling for precipitation when calculating the correlation between τ and temperature (and vice versa).

### 3.5 Implications and future perspective

Our quantitative maps of SOC turnover times in top- and subsoil layers based on an extensive soil profile dataset represents a key step towards a better understanding of global soil carbon stocks and dynamics. Our research illustrates how global variations in τ for both top- and subsoil layers are influenced by the interplay of climatic and edaphic factors, and demonstrates their pronounced heterogeneity within and across biogeographic zones. These analyses revealed complex interactions between temperature, water availability and soil physio-chemical properties, thereby enriching our comprehension of the complex mechanisms driving spatial variability and nonlinearity in τ–environment relationships from

global to local scales. The distinct factors driving $\tau$ in topsoil versus subsoil underscore the importance of incorporating soil depth when assessing large-scale $\tau$ patterns.

The findings in this study also have the potential to improve the parameterization and future projections of ESMs by integrating more accurate global $\tau$ data. Our results support several previous studies that also identified an underestimation of SOC turnover times in ESMs through $^{14}$C observations (He et al., 2016; Shi et al., 2020). This discrepancy between data- and model-derived $\tau$ emphasizes the need to incorporate more detailed edaphic-climate dependent and depth-specific $\tau$ estimates into biogeochemical models to enhance predictive accuracy. Furthermore, there is evidence that the most exchange of soil

carbon with the atmosphere occurs through relatively small and fast soil carbon pools on a short timescale, which can result in a "leaky sink" response when carbon input is elevated (Bradford, 2017; van Groenigen et al., 2017). This response may conceal the longer turnover times in inert pools that are not precisely captured by models. Adjusting turnover rates and carbon transfer parameters in ESMs to align with the longer $\tau$ values we report here may extend the turnover time of "slow" or "passive" pools in models (especially for deep layers) that constitute the majority of soil carbon (Torn et al., 2009; He et

al., 2016).

      To improve the accuracy of carbon cycle models and enhance the projections of future soil carbon sequestration rate and magnitude, it is essential to incorporate edaphic-climatic dependent and depth-resolved estimates of turnover times into these models. These insights across various climate zones, biomes, terrains, and soil properties contribute to reducing uncertainties related to context-dependent effects governing soil carbon stocks and dynamics, thus helping to inform

strategies that enhance sustainable soil management and mitigate the impacts of climate change.

## 4 Data availability

The global maps of top- and subsoil organic carbon turnover times are available online at:
https://doi.org/10.5281/zenodo.14560239 (Zhang, 2025).

## 5 Code availability

The code used for this study is publicly available at: https://github.com/leizhang-geo/global_soil_carbon_turnover_time.git.

## 6 Conclusions

This study provides a comprehensive assessment of global apparent SOC turnover times in both top- and subsoil layers. By integrating state-of-the-art datasets of soil profiles, plant roots, satellite observations, we employed machine learning models to produce the spatially-explicit maps of $\tau$ with quantified uncertainties. The results reveal pronounced spatial heterogeneity

and non-linearity in $\tau$–environment relationships within and among different biomes and climatic zones. Our findings

demonstrate the context-dependent effects of temperature and water availability on τ, which vary depending on soil attributes, mainly including soil texture, carbon-to-nitrogen ratio, cation exchange capacity and soil pH. Considering the potentially large differences in the driving factors of τ from large to small scales, we further mapped the dominating factors of τ in two soil layers at local scales. Overall, this study synthesizes the multiple observation-based datasets that currently available, and provides new global maps of top- and subsoil organic carbon turnover times. The dataset and new insights of this study are expected to serve as a foundation for benchmarking biogeochemical models and supporting effective carbon management.

**Author contributions**

L.Z. and L.Y. conceived the research; L.Z. collected the data and performed the analyses; L.Z., L.Y., T.W.C., C.M.Z., and C.Z. interpreted the analytical results with reviews from all authors; S.D. contributed to the validation and comparison with radiocarbon measurements; G.B.M.H., A.M.J.-C.W., A.-X.Z. and C.Z. contributed to global mapping and uncertainty analysis. L.Z., Y.P., and F.S. contributed to data curation and validation. L.Z., L.Y., T.W.C., and C.M.Z. wrote the paper with substantial contributions from all of the authors.

**Competing interests**

The authors declare that they have no conflict of interest.

**Acknowledgements**

The authors express sincere gratitude to Weimin Ju (International Institute for Earth System Science at Nanjing University), Nuno Carvalhais (Max Planck Institute for Biogeochemistry), Alison M. Hoyt (Earth System Science at Stanford University) and Julian Helfenstein (Soil Geography and Landscape Group at Wageningen University), whose valuable comments and suggestions on an earlier version of our manuscript.

**Financial support**

This work was supported by the National Natural Science Foundation of China (42471468), the Fundamental Research Funds for the Central Universities (0209-14380115), the Leading Funds for the First-Class Universities (020914912203 and 020914902302), and the Research Funds for the Frontiers Science Center for Critical Earth Material Cycling, Nanjing University. T.W.C acknowledges support from DOB Ecology and Bernina Foundation. C.M.Z was funded by the Ambizione grant PZ00P3_193646. L.Z. acknowledges the support from the U.S. Department of Energy, Office of Science, Office of Biological and Environmental Research, under Award Number DE-AC02-05CH11231.

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
