# Peer review of "Mapping global distributions, environmental controls, and uncertainties of apparent top- and subsoil organic carbon turnover times"

_Earth System Science Data, 2024_

## Author Comment (AC1)

**Response to Reviewer 1**

This study creates global maps of soil organic carbon turnover times (τ) at both topsoil and subsoil depths with observation-based datasets and machine learning methods. It addresses an important knowledge gap regarding the global maps related to the depth-dependent variations and environmental controls of soil carbon turnover, which is essential for understanding terrestrial carbon storage and dynamics, especially in the context of climate change. The methods used are well-suited to the scale and complexity of the question being addressed. The use of over ninety thousand geo-referenced soil profiles, satellite-derived environmental variables, and machine learning models, provides a robust framework for mapping and analyzing τ across a wide range of environmental conditions and different biomes. The authors' approach to quantify the uncertainty of their maps also adds considerable value, making their results more applicable to future carbon cycle modelling and land management efforts. In conclusion, I recommend the acceptance of this manuscript after revisions following several suggestions I listed below.

AC: Thank you for your kind words. We are delighted that you acknowledge the importance of our work and recognize that the findings in our work could bridge the gap and help producing a qualified τ map dataset.

(1) Organic carbon turnover times can be calculated from influencing factors, such as carbon allocation belowground, SOC stocks. Why not first generate the spatial distribution data of those factors based on sampling data with machine learning method, then calculate organic carbon turnover times with the physical equations.

AC: Our approach prioritizes direct estimation of τ using a combination of soil profile observations, root distribution data, and machine learning-based spatial predictions. However, we acknowledge the alternative approach suggested by the reviewer — first predicting the spatial distribution of influencing factors (e.g., carbon input and SOC stocks) and then computing τ using physical equations. We chose our approach for the following reasons:

1) Uncertainty propagation and data constraints: many of the influencing factors (e.g., belowground carbon allocation, root distribution) have limited global observations and high spatial variability. If we were to separately predict these variables before calculating τ, errors from each step would compound, potentially increasing overall uncertainty. By training the machine learning model directly at sample locations, we reduce the risk of error propagation from multiple independent modeling steps.

2) Machine learning model ability to capture complex relationships: The turnover time of SOC is influenced by nonlinear interactions between climate, soil properties, and vegetation. Machine learning can capture the feature importance and their interactions effectively from modelling τ– environment relationships at sample locations. In contrast, if using all grid values to fit a model and then analyze the driving mechanisms, attributing variations in τ to multiple environmental variables suffer from a circularity issue.

Therefore, we reckon that using the independent soil sample data to calculate τ at sample locations firstly, and then using a model to make the global τ maps should be a more reasonable choice from both mapping and driving analysis aspects.

(2) From the physical equations 1 to 8, the importance of different factors could be derived. Do the importance results of RF coincide with those equations?

AC: After using those equations to calculate $\tau$ at sample locations, we used RF model to analyze the driving factors (e.g., feature importance) on $\tau$ at global, biome, and local scales. The above two steps have different purpose. The RF model was not used to directly compute $\tau$ but rather assesses how various environmental factors (we focused on climate, soil physio-chemical properties, and topography) correlate with the observed spatial variation in $\tau$.

(3) While the methods section is thorough, some readers may benefit from further clarification on the choice of specific methods, particularly regarding the machine learning model calibration. A more explicit discussion of how the model's hyper-parameters were tuned would help readers better appreciate the methodology's rigor.

AC: We appreciate the reviewer's suggestion to clarify the details of our machine learning model calibration, particularly regarding hyper-parameter selection. When using RF model, there are two important user-defined parameters in the RF. The first is the number of covariates that randomly selected for each tree building process. We used the rounded down square root of the total number of covariates as this parameter value by default (Breiman, 2001). The second parameter is *ntree*, which is defined as the number of trees to be learned in the forest. We set *ntree* = 200, for the previous soil mapping studies showed that it is sufficient to obtain stable results when the number of trees is larger than 150 (Lopes, 2015; Wadoux et al., 2019; Zhang et al., 2021). We added some sentences in Section 2.2 to describe above content.

References:

Lopes, M.E., 2015. Measuring the algorithmic convergence of random forests via bootstrap extrapolation. Technical Report. Department of Statistics. University of California, Davis CA.

Wadoux, A.M.J.C., Brus, D.J., Heuvelink, G.B.M., 2019. Sampling design optimization for soil mapping with random forest. Geoderma 355, 113913. https://doi.org/10.1016/j.geoderma.2019.113913.

Zhang, L., Yang, L., Ma, T., Shen, F., Cai, Y., Zhou, C., 2021. A self-training semi-supervised machine learning method for predictive mapping of soil classes with limited sample data. Geoderma 384, 114809. https://doi.org/10.1016/j.geoderma.2020.114809

(4) The cross-validation procedure is well-executed, but it would be helpful to provide more specific details on how the biome-specific samples were handled during the 10-fold validation process. Ensuring that the sample division properly accounts for the geographical and ecological variation across biomes could further strengthen the model's credibility.

AC: We appreciate the reviewer's request for further clarification on how biome-specific samples were handled during the 10-fold cross-validation process. To ensure that our validation approach accounted for geographical and ecological variation across biomes, we implemented a biome-stratified cross-validation strategy as follows: We stratified the dataset based on biome classifications before performing cross-validation. Each biome was treated as a separate category to ensure that the sample distribution was maintained across all folds. Instead of randomly splitting the entire dataset, we ensured that each fold contained a proportional representation of samples from

each biome. This approach prevented certain biomes (e.g., tundra, boreal forests) from being underrepresented in the validation sets, thereby improving model generalization across diverse ecosystems. When calibrating the model, the method of over-sampling was adopted to increase training data in biomes with smaller sample size, this way can alleviate the imbalance in data when training the model (Chawla, 2010). We clarified these details in the revised manuscript.

Reference:

Chawla, N.V., 2010. Data Mining for Imbalanced Datasets: An Overview, in: Maimon, O., Rokach, L. (Eds.), Data Mining and Knowledge Discovery Handbook. Springer US, Boston, MA, pp. 875–886. https://doi.org/10.1007/978-0-387-09823-4_45

(5) The paper requires careful attention to language for clarity and readability. I would recommend reviewing some descriptions for possible simplifications in sentence structure and corrections in grammar. Some examples are listed below.

18: "… is central to …" -> "… plays a crucial role in …".

AC: Corrected. Thank you.

19: "remain" -> "remains".

AC: Corrected. Thank you.

92: it is better to change the word "minimize" to "reduce".

AC: Corrected. Thank you.

103: "layers observations" -> "layer observations".

AC: Corrected. Thank you.

465: "insights of this study -> "insights from this study".

AC: Corrected. Thank you.

We further made a great effort in polishing the paper so that it is more explicit and clearer.

---

## Author Comment (AC2)

**Response to Reviewer 2**

General comments:

This manuscript calculated the apparent turnover time of top and subsoil SOC on a global scale. The major outcome from this work is very useful and timely needed for soil biogeochemistry and carbon cycle modeling communities. The comprehensive data inputs authors used, random forest based geospatial predictive mapping and in-depth uncertainty analysis guaranteed the quality of the produced $\tau$ map. Overall, this is a solid and interesting work and I would like to recommend it for acceptance after some technical revisions.

AC: Thank you for your kind words. We are delighted that you acknowledge the importance of our work and thank you for your positive conclusion on our manuscript.

Specific comments:

The description of quantile regression forest is not very straightforward. Can authors double check and revise this part?

AC: Thank you for your suggestion. The quantile regression forests (QRF) model estimates the quantiles of the conditional distribution of the target variable at prediction points. For example, the 0.05 and 0.95 quantiles are typically derived to represent the lower and upper limits of a symmetric 90% prediction interval. QRF first constructs a RF model in the usual way, by developing multiple decision trees that use subsets of the training data, whereby the prediction of each tree equals the average of the observations in the end node of the tree in which the prediction point sits. The RF prediction is the average of all tree predictions. Since averaging is a linear process, the RF prediction boils down to a weighted sum over the $n$ observations of the response variable:

$$\hat{y}(x) = \sum_{i=1}^{n} w_i(x) \cdot y_i$$

In QRF, the weights $w_i(x)$ are used to estimate the cumulative distribution $F(y|x) = P(Y \leq y|x)$ of the response variable $Y$, given the covariate data $x$, as follows:

$$\hat{F}(y|x) = \sum_{i=1}^{n} w_i(x) \cdot 1_{\{y_i \leq y\}}$$

where $1_{\{y_i \leq y\}}$ is the indicator function (i.e., it is 1 if the condition is true and 0 otherwise). Any quantile $q$ of the distribution can then be derived by iterating towards the value of $y$ for which $\hat{F}(y|x) = q$ (Meinshausen, 2006).

We added the above in Section 2.4.2 of the revised manuscript to make the description of QRF clearer.

References:

Meinshausen, N.: Quantile Regression Forests, Journal of Machine Learning Research, 7, 983–999, 2006.

There lacks certain discussion of the topographical effect on $\tau$. Since the topography largely impacts

τ in tundra covered regions (Fig. S23, S24), can authors discuss this finding?

AC: We appreciate the reviewer's insightful suggestion. In the main text, we extracted top six important covariates to analyze their directional effects on topsoil and subsoil τ, according to the feature importance results by random forest model at the global scale (Fig. S23a, S24a). Although we hope to focus on describing these six key factors' effects on τ at the global scale in Section 3.3.1, we acknowledge that the effects of topographical factors should also be explained here. In the revised manuscript, at the end of Section 3.3.1, we added a paragraph to discuss it, and highlighted the large impacts of topography on τ in tundra areas and its potential reasons.

Technical corrections:

I suggest authors double check and add units to all variables in your equations.

AC: Thanks for pointing this out. We added the units of SOC content after showing Eq. 1.

Line 89: "collected form" shall be "collected from"

AC: Corrected. Thanks.

Line 99 - 100: In equation 1, SOC_Du-Dl is not defined.

AC: Thanks. We added the definition of this variable in Eq. 1 in the revised manuscript.

Line 100: "Were" shall be "Where"

AC: Corrected. Thanks.

Line 103: "fit layers observations at different depth intervals" can be simplified "fit observations along depth"?

AC: Corrected. Thanks.

Line 105: "the SOCS at two layers for" shall be more specific "the SOCS at top- and subsoil layers for"

AC: Corrected. Thanks.

Line 112: "The flux of carbon at a certain soil layer needs to be obtained through" can be revised "Carbon influx at each soil layer comes from"

AC: Corrected. Thanks.

Line 114: "The annual NPP (kg C m-2 yr-1) produced by the moderate-resolution imaging spectroradiometer (MODIS)" means the MOD17 products? Please add the specific version of the MOD17 product and the url where the author downloaded this data.

AC: Thanks for the suggestion. We added the specific version of this data product and the URL in the revised manuscript. We also provided the reference (Running and Zhao, 2019) (in the main text and Table S1) related to this dataset when describing it.

Reference:

Running, S. and Zhao, M.: MOD17A3HGF MODIS/Terra Net Primary Production Gap-Filled Yearly L4 Global

500 m SIN Grid V006, https://doi.org/10.5067/MODIS/MOD17A3HGF.006, 2019.

Line 126: "total amount of roots" shall be "total root biomass"?
AC: Corrected. Thanks.

Line 132: " the root distribution" shall be " the root biomass distribution"?
AC: Corrected. Thanks.

Line 144: " for each soil sample site, root profile observations within the same terrestrial ecoregions (Dinerstein et al., 2017) and the same soil type (FAO–Unesco, 1990) as that of the soil sample were selected. The corresponding mean of those selected root observations for each soil sample location were finally collected (Figs. S8 and S9)" can be simplified "we apply the arithmetic mean of fr from root profile observations within the same terrestrial ecoregions (Dinerstein et al., 2017) and soil type (FAO–Unesco, 1990) as soil sample."
AC: We revised this sentence following your suggestion. Thanks.

Line 189: "The partial correlation of each influencing factor was calculated while controlling other factors" at which level? Mean or median?
AC: We appreciate the reviewer's request for clarification. The partial correlation of each influencing factor was calculated while controlling for other factors at the mean level using Pearson's partial correlation analysis. This approach estimates the linear relationship between τ and each environmental factor while holding other variables constant, based on their mean values. To improve clarity, we have added "at the mean level" in this sentence for improving clarity following your suggestion.

Line 201: "and this division was performed on each biome data to ensure that the ten split sets can keep a balance among biomes". If my understanding is correct, do you mean "and samples of each biome in each subset has the same proportion as the whole dataset"?
AC: Yes. We revised this sentence following your suggestion. Thanks.

Line 209: "by replacing observations by indicator transforms". Not sure I understand "indicator transforms". Do you mean replacing the actual observation values by a tag of category? If so I'm not sure why this is needed. Can you provide a simple example?
AC: We appreciate the reviewer's request for clarification. In QRF, the term indicator transforms refers to the approach used to estimate the conditional quantiles of the target variable, rather than categorizing observations. Please see our response to the first comment. The term "indicator transforms" referred to the use of the indicator function. To improve clarity, we have revised related contents in Section 2.4.2 to make the description of QRF method clearer.

Line 213: "has been also" shall be "has also been"
AC: Corrected. Thanks.

 If my understanding is correct, it's nice to consider error propagation and produce a dataset considering this uncertainty information to train quantile regression forests. But in your writing this information is not explicitly conveyed so I feel a bit confused when reading this part. It will be better if authors can explicitly tell readers this information above this paragraph.

AC: We consider the uncertainty mainly comes from two sources of error. The first is the model error, which refers to covariates that do not fully explain the variations in the target variable (i.e., top- and subsoil $\tau$ in this study) and error in the estimation of the model parameters. The second is the measurement error, which represents the difference between the actual and recorded value that related to the calculation of the target variable (i.e., the input variables related to $\tau$ calculation) (van der Westhuizen et al., 2022; Heuvelink and Webster, 2023). Therefore, the first two paragraphs in Section 2.4.2 were written to describe how did we considered these two sources of error. To better let the reader understand the ideas of uncertainty calculation, we added some content at the beginning of the first paragraph in Section 2.4.2, to make a linkage between the first source (using QRF to calculate) and the second source (using error propagation to calculate the uncertainty of inputs) of error.

References:

van der Westhuizen, S., Heuvelink, G.B.M., Hofmeyr, D.P., Poggio, L., 2022. Measurement error-filtered machine learning in digital soil mapping. Spatial Statistics 47, 100572. https://doi.org/10.1016/j.spasta.2021.100572

Heuvelink, G.B.M., Webster, R., 2023. Uncertainty assessment of spatial soil information, in: Encyclopedia of Soils in the Environment. Elsevier, pp. 671–683. https://doi.org/10.1016/B978-0-12-822974-3.00174-9

Line 250: Unfinished sentence "The shaded grey area represents the"

AC: Thank you for your careful review. This is a sentence from leftovers in the working version in the original manuscript. We removed it in our revised version.

Figure 6: is interesting. I guess the y axis title "MAT/MAP effect" means the partial regression coefficient between MAT/MAP and $\tau$? Please add a description in figure caption.

AC: Yes, you are correct. We added a sentence in the figure caption as you suggested.

Figure 6: Another question. I know to completely decorrelate climate and edaphic variables is extremely tough, but I would suggest authors provide a correlation matrix to visualize and identify potential issues with "multicollinearity" between 2 climate and 4 edaphic variables.

AC: Here we plot a heatmap figure to show the pairwise correlation between those 6 covariates. It shows that most of them have a relatively low (< 0.4) Pearson correlation coefficient to each other, except pH and MAP. Therefore, we think multicollinearity issue is not significant here.

[Figure]

Figure | Correlation between two climate and four edaphic variables using Pearson correlation analysis.

Figure 6d shows different responses for topsoil and subsoil. As high CEC enhances adsorption, more nutrients would be available for micro-organism and more mineral-organic compounds are formed, which tends to increase sensitivity of SOCS to temperature. But meanwhile, plants are more productive and may have higher plant carbon stock, thus might increase sensitivity of NPP to temperature. The results might be the combination of both effects.

AC: Thank you for this insightful comment. We agree with your idea and added this discussion in the revised manuscript (Section 3.2.2).

Figure 7c: What is the unit of variable importance (%)?

AC: Yes, the unit is %. We revised the figure to add this unit.

Line 405: "biogeochemical simulations by ESMs, and will be useful to improve" can be simplified to "ESM simulations and improve"

AC: Corrected. Thanks.

Line 407: shall mention CMIP6 outputs from which experiment?

AC: We analyzed historical simulations outputs of selected ESMs from CMIP6. The historical scenario simulations (also known as the 20th-century simulations) for CMIP6 were carried out for the period from the start of the industrial revolution to near present: 1850–2015. We added a sentence in Section 3.4 to mention it.

From Fig. S17, it seems like most of the wetland samples are collected from the tropics. Would you justify the performance of wetland τ the arctic region?

AC: We appreciate the reviewer's observation regarding the distribution of wetland samples. It is true that a significant portion of the wetland data in our study originates from tropical regions, which may influence the model's ability to accurately capture τ variations in arctic wetlands. However, we have taken several steps to mitigate potential biases and ensure reasonable performance in the arctic region. First, our dataset does include arctic wetland observations from sources such as the Northern

Circumpolar Soil Carbon Database (Hugelius et al., 2013) and other high-latitude soil profiles from WoSIS database. These data contribute to model training and validation, particularly for permafrost-affected regions. Second, the model does not rely solely on the spatial distribution of training samples but also incorporates key environmental variables (e.g., temperature, precipitation, soil physio-chemical properties, and topography) that distinguish arctic from tropical wetlands. This helps the model extrapolate τ values in arctic regions based on known biogeochemical relationships rather than direct spatial proximity. Finally, we extracted our modelling result in the arctic wetland region, and showed the validation accuracy in topsoil and subsoil as follow. Generally, the we obtained an acceptable accuracy on validation set, which justifies the performance of τ predictions in this region.

[Figure]

Figure | Validation plots for predictions of soil organic carbon turnover time (τ, yr) in top- (a) and subsoil (b) layer.

References:

Hugelius, G., Tarnocai, C., Broll, G., Canadell, J. G., Kuhry, P., and Swanson, D. K.: The Northern Circumpolar Soil Carbon Database: spatially distributed datasets of soil coverage and soil carbon storage in the northern permafrost regions, Earth System Science Data, 5, 3–13, https://doi.org/10.5194/essd-5-3-2013, 2013.

---

## Author Comment (AC3)

**Response to Reviewer 3**

The manuscript titled "Mapping global distributions, environmental controls, and uncertainties of apparent top- and subsoil organic carbon turnover times" addresses an important knowledge gap and is therefore likely to be of broad interest. The primary aim of the study is to map carbon turnover times for the topsoil (0–30 cm) and subsoil (30–100 cm) at a global scale, with a spatial resolution of 30 arcsec. The resulting maps, along with their associated prediction uncertainties, are freely available on Zenodo.

Overall, I am satisfied with the manuscript and the dataset provided on Zenodo. However, I recommend a few minor revisions before it is suitable for publication. These are as follows:

AC: Thank you for your positive general comment. Our responses to your specific comments as below.

(1) A few minor spelling and grammatical corrections are needed to enhance the text's readability.

AC: Thank you for pointing this out. We have revised the manuscript to correct these errors in language.

(2) I am curious why the bias (mean error) was not calculated. Demonstrating that the maps are unbiased would strengthen the validity of the results.

AC: We appreciate the reviewer's suggestion regarding the calculation of bias (mean error) in addition to RMSE. In our study, we primarily used RMSE (root mean squared error) as the validation metric, as it accounts for both the variance of errors and their magnitude, making it a robust measure of prediction accuracy. However, we acknowledge that assessing bias (mean error) (here we think the metric name should be MAE [mean absolute error]) would provide additional insight into systematic over- or under-predictions. To address this, we have now calculated the mean error (ME) alongside RMSE and included these results in the supplementary material (Table S3, also show as below). The MAE values confirm that the predicted $\tau$ values do not exhibit significant systematic bias, further supporting the validity of our mapping approach.

**Table. Global cross-validation results for top- and subsoil $\tau$ predictions.**

| Variable to predict | RMSE | MAE | $R^2$ | MEC |
|---|---|---|---|---|
| Topsoil $\tau$ | 14.60 | 5.72 | 0.87 | 0.93 |
| Subsoil $\tau$ | 175.05 | 65.57 | 0.70 | 0.83 |

Note: RMSE, root mean squared error; MAE, mean absolute error; $R^2$, coefficient of determination; MEC, model efficiency coefficient.

(3) I am also wondering why the R-squared value was used instead of the model efficiency coefficient (MEC) (https://en.wikipedia.org/wiki/Nash%E2%80%93Sutcliffe_model_efficiency_coefficient). MEC is often considered more appropriate than R-squared in similar contexts.

AC: We appreciate the reviewer's suggestion regarding the use of the model efficiency coefficient

(MEC) as an alternative to $R^2$. In our study, we chose $R^2$ as a measure of model performance because it is widely used in machine learning-based predictive mapping to assess the proportion of variance explained by the model. Additionally, $R^2$ allows for direct comparison with previous studies that have employed similar modeling approaches. We acknowledge that MEC is also particularly useful in environmental modelling and soil mapping contexts. Thus, to further strengthen our validation, we have now computed MEC alongside $R^2$, and included these results in the supplementary material (Table S3). The MEC values are generally consistent with $R^2$, reinforcing the reliability of our predictions.

(4) Clarification of Section 2.4.2 (Quantification of uncertainty): I believe this section could be revised to provide clearer and more reproducible methodological details. If I understand correctly, did you fit a single QRF model using approximately 90,000 × 100 carbon turnover observations to account for input uncertainties? Or did you fit 100 different QRF models and then combine the results? Clarifying this would enhance transparency.

AC: In our study, we considered the potential error (represented by the calculated standard deviation [SD] input variables) in estimating $\tau$ for each sample point. Then, a Monte Carlo approach was adopted to incorporate the SD in the estimated $\tau$ at each sample. That is, the value of $\tau$ at each sample location was randomly drawn 100 times from a normal distribution given the known mean and SD. Therefore, we generated 95,200 × 100 and 66,807 × 100 samples for topsoil and subsoil, respectively. Finally, we used these samples to fit a QRF model, and the model can directly produce the 0.05 quantile and 0.95 quantile of predictions so that helped us obtain uncertainty maps. We revised some sentences in Section 2.4.2 to make this description clearer.

(5) It would be beneficial to include guidance on how to interpret the PIR values. This would help readers better assess the quality of the maps and understand the magnitude of the prediction uncertainty.

AC: Yes, we agree to improve the explanation of prediction interval ratio (PIR) in the manuscript to help readers have a better understanding. PIR provides a relative measure of uncertainty by comparing the width of the prediction interval to the predicted median value. A lower PIR indicates higher confidence in the predictions, while a higher PIR suggests greater uncertainty. To improve clarity, we added the guidance on how to interpret PIR values in the revised manuscript.

(6) Similarly, a more detailed explanation of PICP would be helpful. Many readers may be unfamiliar with this metric, including what it measures, how to interpret it, and how it reflects the fairness of the uncertainty quantifications.

AC: We agree to add explanation of prediction interval coverage probability (PICP) would be helpful, as many readers might be unfamiliar with it. PICP is a metric used for evaluating the reliability of uncertainty quantifications by measuring the proportion of observed values that fall within a given prediction interval. It can effectively evaluate if the probability assigned to the prediction intervals is equal to the frequency of empirical test data within the prediction intervals (Goovaerts, 2001; Malone et al., 2011; Schmidinger and Heuvelink, 2023) To improve clarity, we have expanded the explanation of PICP in the revised manuscript. We clarify that: PICP is defined

as the fraction of observed values that fall within the predicted uncertainty bounds (e.g., the 90% prediction interval). A PICP of 0.90 indicates that 90% of observed values lie within the computed prediction interval, suggesting well estimated uncertainty. If PICP is significantly lower than the expected interval probability (e.g., < 0.90 for a 90% interval), it implies that the model underestimates uncertainty, potentially leading to overconfidence in predictions. Conversely, if PICP is much higher than the expected probability (e.g., > 0.90 for a 90% interval), it suggests overestimated uncertainty, meaning the prediction intervals may be too wide. We also put more detailed explanation into the caption of Figure S22 in supplementary material.

References:

Goovaerts, P.: Geostatistical modelling of uncertainty in soil science, Geoderma, 103, 3–26, https://doi.org/10.1016/S0016-7061(01)00067-2, 2001.

Malone, B. P., McBratney, A. B., and Minasny, B.: Empirical estimates of uncertainty for mapping continuous depth functions of soil attributes, Geoderma, 160, 614–626, https://doi.org/10.1016/j.geoderma.2010.11.013, 2011.

Schmidinger, J. and Heuvelink, G. B. M.: Validation of uncertainty predictions in digital soil mapping, Geoderma, 437, 116585, https://doi.org/10.1016/j.geoderma.2023.116585, 2023.

(7) The sections on mapping and uncertainty quantification appear somewhat intertwined. Did you first fit an RF model to create the maps and then use a separate QRF model (including the inputs' uncertainty) to quantify the uncertainty associated with the RF predictions? Did you fix the (pseudo-)random number generator to maintain consistency between the RF and QRF models? Clarifying these aspects would enhance the transparency and reproducibility of the methodology.

AC: We first trained a standard RF model to generate the spatially explicit predictions of $\tau$. Then, a QRF model was adopted for uncertainty quantification. To maintain consistency between the RF and QRF models, the QRF model was trained using the datasets. We also fixed the pseudo-random number generator during the training procedure for these two models. This ensures that both models operated with the same data splits and bootstrapped samples, minimizing variations introduced by randomness and enhancing reproducibility. To improve clarity, we have added this description in Section 2.4.2.

Once these minor issues are addressed, I believe the manuscript will be ready for publication.

AC: We thanks the reviewer for this valuable suggestion above, which has helped us enhance the quality of our manuscript.